# The Longitudinal Plasma Modes of *κ*-Deformed Kaniadakis Distributed Plasmas Carrying Orbital Angular Momentum

**DOI:** 10.3390/e24091211

**Published:** 2022-08-29

**Authors:** Ling Tan, Qiaoyun Yang, Hui Chen, Sanqiu Liu

**Affiliations:** Jiangxi Province Key Laboratory of Fusion and Information Control, Department of Physics, Nanchang University, Nanchang 330031, China

**Keywords:** orbital angular momentum, Langmuir waves, ion-acoustic waves, κ-deformed Kaniadakis distribution function, 52.20.-j, 52.25.Dg, 52.35.Fp

## Abstract

Based on plasma kinetic theory, the dispersion and Landau damping of Langmuir and ion-acoustic waves carrying finite orbital angular momentum (OAM) were investigated in the κ-deformed Kaniadakis distributed plasma system. The results showed that the peculiarities of the investigated subjects relied on the deformation parameter κ and OAM parameter η. For both Langmuir and ion-acoustic waves, dispersion was enhanced with increased κ, while the Landau damping was suppressed. Conversely, both the dispersion and Landau damping were depressed by OAM. Moreover, the results coincided with the straight propagating plane waves in a Maxwellian plasma system when κ=0 and η→∞. It was expected that the present results would give more insight into the trapping and transportation of plasma particles and energy.

## 1. Introduction

In 1990, Tamm et al. produced a Laguerre–Gaussian (LG)-mode laser beam with helical wave fronts that can drive molecules and neutral atoms [1]. Allen et al. subsequently demonstrated that the angular momentum carried by a laser beam with azimuth phase distribution was unrelated to the state of the polarized photons, and that the missing parts of the angular momentum of photons are to be found in twisted electromagnetic beams or optical vortices [2]. It is well recognized that the angular momentum of electromagnetic radiation has two distinct components. The first is the intrinsic part associated with wave polarization, or spin; the second is an extrinsic part related to the orbital angular momentum (OAM) that depends on the spatial radiation distribution [3]. Hence, the laser beams, as depicted by the LG function that satisfies the basic orthogonal condition, possess spin and angular momentum as well as OAM, which lays the foundation for numerous important scientific applications because of its inherent orthogonality and production techniques that have matured in the laboratory [4,5,6,7,8]. For example, owing to the special helical phase and hollow light field of the OAM beam, OAM can be applied to micro-control technologies in the microscopic world, such as optical tweezers and micromotors [9]. As an independent degree of freedom for wireless and quantum communications, OAM can be used to achieve a higher communication capacity through the simultaneous transmission of multiple orthogonal OAM mode vortex beams [10,11,12,13]. It has even been suggested that telescopes equipped with OAM diagnostic instruments can be made to detect rotating black holes [14,15].

The energy, momentum and angular momentum of laser beams can be transferred to matter by interacting with it. Moreover, the propagation of an OAM beam in plasma is associated with the excitation of a plasma wave, which may likewise carry OAM. In recent years, related research carried out on OAM in plasma has attracted much attention, and the significance of collective plasma oscillations with OAM has been recognized in different contexts. For instance, Mendonça et al. first employed the concepts of photon OAM states in plasma systems to investigate the stimulated Raman and Brillouin backscattering of collimated beams [16]. It was shown that the exchange of OAM between electromagnetic and electrostatic waves occurred in stimulated Raman and Brillouin backscattering, which implied that plasmon and phonon states carried OAM. The idea opened the door for various studies on OAM in plasma. By using two fluid models and Ampere’s law in a magnetized plasma, Shukla indicated that three-dimensional modified-kinetic Alfvén waves can propagate in the shape of Alfvénic tornadoes featuring plasma density whirls or magnetic flux ropes with OAM [17]. Vieira et al. confirmed the existence of LG modes in particle-in-cell (PIC) simulations of intense laser–plasma interactions, and revealed the mechanism of high OAM harmonic generation and amplification through stimulated Raman backscattering in plasma [18,19]. Ali et al. calculated the quasistatic axial magnetic field generated during laser propagation in plasma by taking into account both the spin and OAM of the laser pulse [20,21]. The fluid theory of electron-acoustic waves in a two-temperature electron plasma was considered by Shahzad et al., who derived the OAM density of electron-acoustic waves [22]. Ali et al. subsequently presented dust oscillons with distinct OAM states in a collisionless unmagnetized self-gravitating dusty plasma [23].

The OAM modes in plasma also introduced other significant effects such as single-electron level twisted photon emission [24], toroidal shaped plasma turbulence in radio-pumping [25], and a helical plasma accelerator [26].

Moreover, the quasistatic axial magnetic field generated by OAM beams in plasma also has prospective applications in deep resource exploration, atmospheric science, and underwater communications [27].

In the study of the interplay between light beams and plasma, wave-particle interaction plays a pivotal role in particle acceleration, wave mixing, and the nonlinear decay of laser beams with OAM, for which a kinetic framework is necessary. Relying on the plasma kinetic description, Mendonça derived the dispersion relation and Landau damping of helical electron plasma waves with OAM in cylindrical geometry under the paraxial approximation. It was shown that the vertical velocity component of a helical Langmuir wave also contributes to Landau resonance [28,29]. Following the work of Mendonça, Khan et al. extended the theory to ion-acoustic plasma vortices with OAM and indicated that the azimuthal component of an electric field produces optical torque on the medium, which results in increased OAM of the plasma vortex [30]. Rehman et al. studied the propagation characteristics of an electronic acoustic wave in a two-electron component plasma. The results showed that the electronic acoustic wave carrying OAM was strongly damped at large and intermediate wavelengths, whereas it was weakly damped at small wavelengths [31]. Recently, Khan et al. described the helical structure of electrostatic plasma waves carrying OAM by introducing a variable transformation. The proposed idea improved the method of accessing wave damping [32]. The kinetic theory for these OAM-carrying plasma waves were investigated in a Maxwellian distributed plasma system. However, during solar wind or flares, pulsars and other complex environments, plasma systems with superheated electrons exhibit energetic tails in particle velocity or energy distribution; therefore, the Maxwellian distribution is not applicable [33].

To process some emerging physical problems in complex environments, attempts were made to generalize statistical mechanics on conventional Boltzmann–Gibbs (BG) entropy. In this context, Rényi proposed the non-extensive generalization of BG entropy [34], which was later also suggested by Tsallis [35], whose non-extensive entropy was in excellent agreement [36] with experimental data [37,38]. Afterwards, Kaniadakis put forward a new so-called κ-deformed distribution in 2001 [39], which has been widely applied to the kinetics of interaction atoms and photons [40], nonlinear kinetics [41,42,43], cosmic rays [33], blackbody radiation [44], quantum entanglement [45], quark–gluon plasma formation [46], and even financial systems [47,48] and epidemiology [49]. The κ-deformed distribution arising from Kaniadakis entropy covers both nonextensive and the classical Maxwell–Boltzman distributions [39]. In subsequent studies, Beck and Cohen proposed that this κ-deformed distribution can be regarded as the result of more generalized statistics known as superstatistics [50]. Ourabah et al. also verified that the nonthermal and suprathermal empirical distributions can be recovered from Beck–Cohen superstatistics [51]. The κ-deformed distribution can be represented as a more universal form of the distribution functions mentioned in the above work. Consequently, many studies have been revisited under the κ-deformed distribution, such as longitudinal plasma modes [52] and Jeans gravitational instability [53,54,55]. The κ-deformed distribution as a generalized statistic may be able to characterize plasma waves in a fusion device. The power density of the OAM beam highly localized away from the propagation axis could be an efficient device for transferring concentrated heating power, which could be used to heat the fusion plasma at certain locations. The OAM state may also act as a potential plasma diagnostic technique since it can be modulated by various anisotropic and nonuniform structures in plasma [56].

To explore the effects of OAM and the κ-deformed Kaniadakis distribution function on the dispersion and Landau damping of longitudinal plasma waves in this paper, longitudinal plasma modes carrying OAM were considered in κ-deformed Kaniadakis distributed plasmas based on kinetic theory. This manuscript is organized in the following fashion. In Section 2, linear kinetic theory is employed to derive a dielectric function for the longitudinal plasma waves in κ-deformed Kaniadakis distributed plasmas. Section 3 and Section 4 describe the linear dispersion relation and the damping rate of Langmuir and ion-acoustic waves, respectively. The numerical results and a brief summary are given in Section 5.

## 2. The Longitudinal Dielectric Function

According to plasma kinetic theory, the dispersion relation and Landau damping of electrostatic waves in an unmagnetized collisionless isotropic plasma are determined by
(1)Reεlω,k=0,
and
(2)γω,k=−Imεlω,k∂/∂ωReεlω,k,
respectively, where Reεlω,k and Imεlω,k are real and imaginary parts of the plasma longitudinal dielectric function $\varepsilon_{l}\left(\omega ,k\right)$. To investigate the novel properties of finite OAM carried by electrostatic waves in a κ-deformed Kaniadakis distributed plasma system, the LG function is used to describe the perturbed electrostatic potential and distribution function, which followed the same method as mentioned in Refs. [29,30,31]. Then the dielectric function was obtained by the linearized Vlasov–Poisson equation [28] given by
(3)$$\varepsilon_{l}\left(\omega ,q_{eff}\right)=1+\sum_{\alpha }\frac{\omega_{p\alpha }^{2}}{k^{2}}\int \frac{q_{eff}\cdot \partial f_{\alpha }/\partial v\;}{\omega -q_{eff}\cdot v}dv.$$
Here, $\omega_{p\alpha }$ is the plasma frequency; α represents plasma species (α=i for ion, α=e for electron, respectively); and $q_{eff}=-iq_{r}{\hat{e}}_{r}+lq_{\theta }{\hat{e}}_{\theta }+\left(k-iq_{z}\right){\hat{e}}_{z}$. More details about the relevant parameters are given in Refs. [29,30,31]). The κ-deformed Kaniadakis distribution function is written as [39]
(4)$$f_{\alpha }\left(v\right)=A_{\kappa }\exp_{\kappa }\left(-\frac{v^{4}}{4\tau^{4}}\right),$$
with
(5)$$\exp_{\kappa }\left(x\right)={\left(\sqrt{1+\kappa^{2}x^{2}}+\kappa x\right)}^{\frac{1}{\kappa }},$$
where $\tau =k_{B}T_{\alpha }/m_{\alpha }$ in relation to the thermal velocity $v_{T\alpha }=\sqrt{k_{B}T_{\alpha }/m_{\alpha }\;}$; and $A_{{}_{\kappa }}$ is the normalized constant given by
(6)$$A_{k}=\frac{n_{\alpha 0}}{{\left(2\pi \right)}^{\frac{3}{2}}\tau^{3}}{\left|2\kappa \right|}^{\frac{3}{2}}\left(1+\frac{3}{4}\left|2\kappa \right|\right)\frac{\Gamma \left(\frac{1}{\left|2\kappa \right|}+\frac{3}{4}\right)}{\Gamma \left(\frac{1}{\left|2\kappa \right|}-\frac{3}{4}\right)}.$$
Compared to the Maxwellian case, the effective temperature of the superstatistics depends on the deformation parameter κ, which was explicitly given in Ref. [53], as
(7)$$T_{\alpha \left(eff\right)}=\frac{1}{\left|2\kappa \right|}\frac{1+\frac{3}{4}\left|2\kappa \right|}{1+\frac{5}{4}\left|2\kappa \right|}\frac{\Gamma \left(\frac{1}{\left|2\kappa \right|}-\frac{5}{4}\right)}{\Gamma \left(\frac{1}{\left|2\kappa \right|}+\frac{5}{4}\right)}\frac{\Gamma \left(\frac{1}{\left|2\kappa \right|}+\frac{3}{4}\right)}{\Gamma \left(\frac{1}{\left|2\kappa \right|}-\frac{3}{4}\right)},$$
In Equation (Equation 6), κ represents the strength of the deformation and the symbol Γ pinpoints the gamma function. To obtain $A_{k}$, the following integral was used [33,57]
(8)$$\int_{0}^{\infty }x^{r-1}\exp_{\kappa }\left(-x\right)dx=\frac{{\left|2\kappa \right|}^{-r}}{1+r\left|\kappa \right|}\frac{\Gamma \left(\frac{1}{2\left|\kappa \right|}-\frac{2}{r}\right)}{\Gamma \left(\frac{1}{2\left|\kappa \right|}+\frac{2}{r}\right)}\Gamma \left(r\right).$$
When κ→0, it is important to note that the Kaniadakis distribution function is reduced to the Maxwellian distribution with $\lim_{\kappa \to 0}\;\exp_{\kappa }\left(x\right)\equiv \exp\left(x\right)$ [33].

The use of the κ-deformed Kaniadakis distribution results in the dielectric function of longitudinal waves carrying OAM which correlated strongly with the azimuthal velocity contribution, as
(9)$$\varepsilon_{l}\left(\omega ,k\right)=1+\sum_{\alpha }\frac{\omega_{p\alpha }^{2}}{C_{k}k^{2}v_{T\alpha }^{2}}\left[B_{k}-Z\left(\xi_{z}^{\alpha }\right)-Z\left(\xi_{\theta }^{\alpha }\right)\right],$$
where
$$B_{k}=\frac{{\left|2\kappa \right|}^{-\frac{1}{2}}\Gamma \left(\frac{1}{\left|2\kappa \right|}-\frac{1}{4}\right)\sqrt{\pi }}{1+\frac{1}{4}\left|2\kappa \right|\Gamma \left(\frac{1}{\left|2\kappa \right|}+\frac{1}{4}\right)},$$
$$C_{k}=\frac{{\left|2\kappa \right|}^{-\frac{3}{2}}\Gamma \left(\frac{1}{\left|2\kappa \right|}-\frac{3}{4}\right)\sqrt{\pi }}{1+\frac{3}{4}\left|2\kappa \right|\Gamma \left(\frac{1}{\left|2\kappa \right|}+\frac{3}{4}\right)},$$
$Z\left(\xi_{s}^{\alpha }\right)$ is the modified plasma dispersion function that includes both axial and azimuthal contributions with $\xi_{z}^{\alpha }=\omega /\sqrt{2}kv_{T\alpha }$ and $\xi_{\theta }^{\alpha }=\omega /\sqrt{2}lq_{\theta }v_{T\alpha }$, respectively, in the presence of the κ-deformed Kaniadakis distribution function, which can be written as
(10)$$Z\left(\xi_{s}^{\alpha }\right)=\int_{-\infty }^{\infty }\frac{\xi_{s}^{\alpha }}{x-\xi_{s}^{\alpha }}{\left(\sqrt{1+\kappa^{2}x^{4}}-\kappa x^{2}\right)}^{\frac{1}{\kappa }}dx.$$

By making use of the Plemelj formula [58] and integrating for Equation (Equation 10), one can obtain the modified dispersion functions under the limitation $\xi_{z,\theta }^{\alpha }\gg 1$ and $\xi_{{}_{z,\theta }}^{\alpha }\ll 1$, as
(11)$$\begin{array}{cc} \; & Z\left(\xi_{z}^{\alpha }\right)+Z\left(\xi_{\theta }^{\alpha }\right)=B_{k}+\frac{1}{2{\left(\xi_{z}^{\alpha }\right)}^{2}}C_{k}+\frac{3}{4{\left(\xi_{z}^{\alpha }\right)}^{4}}D_{k}+\frac{1}{2{\left(\xi_{\theta }^{\alpha }\right)}^{2}}C_{k}+\frac{3}{4{\left(\xi_{\theta }^{\alpha }\right)}^{4}}D_{k}\\ \; & -i\pi \left[\xi_{z}^{\alpha }\exp_{\kappa }\left(-{\left(\xi_{z}^{\alpha }\right)}^{2}\right)+\xi_{\theta }^{\alpha }\exp_{\kappa }\left(-{\left(\xi_{\theta }^{\alpha }\right)}^{2}\right)\right], \end{array}$$
and
(12)$$Z\left(\xi_{z}^{\alpha }\right)+Z\left(\xi_{\theta }^{\alpha }\right)=-i\sqrt{\pi }\xi_{z}^{\alpha }-i\sqrt{\pi }\xi_{\theta }^{\alpha },$$
respectively, with
$$D_{k}=\frac{{\left|2\kappa \right|}^{-\frac{5}{2}}\Gamma \left(\frac{1}{\left|2\kappa \right|}-\frac{5}{4}\right)\sqrt{\pi }}{1+\frac{5}{4}\left|2\kappa \right|\Gamma \left(\frac{1}{\left|2\kappa \right|}+\frac{5}{4}\right)},$$
which were given by Chen [53].

## 3. The Disperation and Landau Damping of Langmuir Waves

It is generally acknowledged that the existence of a Langmuir wave requires that $\frac{\omega }{k}\gg v_{Te}$, namely, $\xi_{z,\theta }^{e}\gg 1$, and then from Equations (9) and (11) the real part of the longitudinal dielectric function for an electron can be given by
(13)Reεleω,k=1−1k2λDe2k2vTe2+lqθ2vTe2ω2−3DkCkωpe2k2k4vTe2+lqθ4vTe2ω4,
which, in combination with Equation (Equation 1), gives rise to the dispersion relation for Langmuir waves with OAM as
(14)$$\omega^{2}=\omega_{pe}^{2}\left(1+\frac{1}{\eta^{2}}\right)+\frac{3D_{k}}{C_{k}}k^{2}v_{Te}^{2}\frac{\left(1+\frac{1}{\eta^{4}}\right)}{\left(1+\frac{1}{\eta^{2}}\right)}.$$
In the above expression, $\eta =\frac{k}{lq_{\theta }}$ is the dimensionless parameter showing the helical phase structure involving the plasma oscillations which are directly associated with OAM. Here, it was obvious that the kinetic dispersion relation was similar to ordinary plane waves in form. Nevertheless, it should be pointed out that the dispersion of a Langmuir wave in an OAM state relies on the deformation parameter κ and OAM parameter η. Moreover, when the azimuthal wave number approached zero (η→∞), Equation (Equation 14) ultimately recovered the following plane wave dispersion relation
(15)$$\omega^{2}=\omega_{pe}^{2}+\frac{3D_{k}}{C_{k}}k^{2}v_{Te}^{2},$$
which was consistent with the results of Langmuir waves in the κ-deformed Kaniadakis distributed plasma system studied by Chen [53].

In the limit κ→0, one has $D_{k}/C_{k}\to 1$, and then Equation (Equation 14) readily reduces to the result of the Langmuir waves with OAM derived in Maxwellian plasma [29],
(16)$$\omega^{2}=\omega_{pe}^{2}\left(1+\frac{1}{\eta^{2}}\right)+3k^{2}v_{Te}^{2}\frac{\left(1+\frac{1}{\eta^{4}}\right)}{\left(1+\frac{1}{\eta^{2}}\right)}.$$
The expression for the imaginary part of the dielectric function was obtained from Equations (9) and (11)
(17)Imεleω,k=πCkk2λDe2ξzeexpκξze2+ξθeexpκξθe2.
Then, the Landau damping for Langmuir waves carrying OAM with Equation (Equation 2) yielded
(18)$$\begin{array}{cc} \; & \\ \; & \gamma =-\frac{\omega^{4}\pi }{2\sqrt{2}C_{k}k^{3}v_{Te}^{3}\left(1+\frac{1}{\eta^{2}}\right)}\left[\exp_{\kappa }\left(-\frac{\omega^{2}}{2k^{2}v_{Te}^{2}}\right)+\eta \exp_{\kappa }\left(-\frac{\omega^{2}\eta^{2}}{2k^{2}v_{Te}^{2}}\right)\right]. \end{array}$$
Here $\lambda_{De}$ is the electron Debye length. Note that the inclusion of the azimuthal velocity component led to the existence of an OAM parameterv η. Equation (Equation 18) shows that the Landau damping of Langmuir waves in a κ-deformed Kaniadakis distributed plasma system was significantly modified by the OAM parameter η and the parameter κ of the distribution function. Again by setting the OAM parameter η→∞, the Landau damping eventually arrived at the simplified expression
(19)$$\gamma =-\frac{\omega^{4}\pi }{2\sqrt{2}C_{k}k^{3}v_{Te}^{3}}\exp_{\kappa }\left(-\frac{\omega^{2}}{2k^{2}v_{Te}^{2}}\right),$$
which was obtained in the absence of the Landau damping OAM in conventional κ-deformed Kaniadakis distributed electron plasma [53].

In Equation (Equation 18), when κ=0, the coefficient $C_{k}\to \pi$, and the Landau damping for Langmuir waves with OAM was reduced to the form of a Maxwellian case [29]:(20)$$\gamma =-\sqrt{\frac{\pi }{8}}\omega_{pe}\frac{\omega_{pe}^{3}}{k^{3}v_{Te}^{3}}\left(1+\frac{1}{\eta^{2}}\right)\left[\exp\left(-\omega^{2}/2k^{2}v_{Te}^{2}\;\right)+\eta \exp\left(-\eta^{2}\omega^{2}/2k^{2}v_{Te}^{2}\right)\right].$$

## 4. The Disperation and Landau Damping of Ion-Acoustic Waves

Under the ion-acoustic time scale $\xi_{z,\theta }^{e}\ll 1$ and $\xi_{z,\theta }^{i}\gg 1$, one has the ion-dielectric function for low-frequency longitudinal modes from Equations (9) and (11),
(21)$$\begin{array}{cc} \; & \varepsilon_{l}^{i}\left(\omega ,k\right)=1-\frac{\omega_{pi}^{2}}{C_{k}k^{2}v_{Ti}^{2}}\left[\frac{C_{k}}{2}\left(\frac{1}{{\left(\xi_{z}^{i}\right)}^{2}}+\frac{1}{{\left(\xi_{\theta }^{i}\right)}^{2}}\right)+\frac{3D_{k}}{4}\left(\frac{1}{{\left(\xi_{z}^{i}\right)}^{4}}+\frac{1}{{\left(\xi_{\theta }^{i}\right)}^{4}}\right)\right]+\\ \; & i\frac{\pi }{C_{k}k^{2}\lambda_{Di}^{2}}\left[\xi_{z}^{i}\exp_{\kappa }\left(-{\left(\xi_{z}^{i}\right)}^{2}\right)+\xi_{\theta }^{i}\exp_{\kappa }\left(-{\left(\xi_{\theta }^{i}\right)}^{2}\right)\right]. \end{array}$$
Substituting Equations (13) and (21) into the relational expression $\varepsilon_{l}\left(\omega ,k\right)=1+\left[\varepsilon_{l}^{e}-1\right]+\left[\varepsilon_{l}^{i}-1\right]$ along with Equation (Equation 1), the dispersion equation yielded
(22)Reεl=1+ωpe2Ckk2vTe2Bk−ωpi2Ckk2vTi2Ck21ξzi2+1ξθi2+3Dk41ξzi4+1ξθi4=0.
Then we arrived at the dispersion relation for ion-acoustic waves carrying OAM, as
(23)$$\omega^{2}=\omega_{pi}^{2}\left(1+\frac{1}{\eta^{2}}\right)\left[\frac{k^{2}\lambda_{De}^{2}\frac{C_{k}}{B_{k}}}{k^{2}\lambda_{De}^{2}\frac{C_{k}}{B_{k}}+1}+\frac{3D_{k}}{C_{k}}k^{2}\lambda_{Di}^{2}\frac{1+\frac{1}{\eta^{4}}}{{\left(1+\frac{1}{\eta^{2}}\right)}^{2}}\right],$$
where $\lambda_{Di}$ is the ion Debye length. In the limit η→∞, $C_{k}/B_{k}\to 1$, and the Equation (Equation 23) was reduced to the same dispersion relation for planar ion-acoustic waves in the κ-deformed Kaniadakis distributed plasma system [53], as
(24)$$\omega^{2}=\omega_{pi}^{2}\frac{k^{2}\lambda_{De}^{2}C_{k}/B_{k}\;}{k^{2}\lambda_{De}^{2}C_{k}/B_{k}\;+1}.$$
In addition, the Maxwellian limit of Equation (Equation 23) can essentially be recovered by setting κ=0:(25)$$\omega^{2}=\omega_{pi}^{2}\left(1+\frac{1}{\eta^{2}}\right)\frac{k^{2}\lambda_{De}^{2}}{k^{2}\lambda_{De}^{2}+1}+3k^{2}\lambda_{Di}^{2}\frac{1+\frac{1}{\eta^{4}}}{1+\frac{1}{\eta^{2}}}.$$
Likewise, according to Equations (2) and (21), the Landau damping for ion-acoustic waves in an OAM state can easily be derived as follows:(26)$$\gamma =\frac{\omega^{4}\pi }{2\sqrt{2}C_{k}k^{3}v_{Ti}^{3}\left(1+\frac{1}{\eta^{2}}\right)}\left[\exp_{\kappa }\left(-\frac{\omega^{2}}{2k^{2}v_{Ti}^{2}}\right)+\eta \exp_{\kappa }\left(-\frac{\omega^{2}\eta^{2}}{2k^{2}v_{Ti}^{2}}\right)\right].$$
Similarly, in Equation (Equation 26), by taking η→∞ and κ=0, the Landau damping for ion-acoustic waves carrying OAM was then reduced to the standard result in a Maxwellian plasma system: (27)$$\gamma =-\sqrt{\frac{\pi }{8}}\omega \frac{\omega^{3}}{k^{3}v_{Te}^{3}}\exp\left(-\omega^{2}/2k^{2}v_{Ti}^{2}\;\right).$$

## 5. Discussion and Conclusions

For graphical illustration, we analyzed the dispersion relation and Landau damping of Langmuir waves and ion-acoustic waves with OAM in a κ-deformed Kaniadakis distributed plasma. How the dispersion and the damping rate of Langmuir waves varied with wave number, for various values of the deformation parameter κ while retaining the OAM parameter η=1 are graphically displayed in Figure 1 and Figure 2.

In Figure 1, the dashed blue, green, purple and solid red curves correspond to κ=0.15, κ=0.20, κ=0.25 and κ=0.35, respectively, and were acquired through solving Equation (Equation 14) numerically, whereas the solid black curve corresponding to κ=0 was obtained by solving the analytically derived Equation (Equation 16). It is clear that as the κ increased, so did the phase velocity, which indicated that increasing the deformation parameter κ enhanced Langmuir wave dispersion.

In Figure 2, the dashed blue, green, purple and solid red curves, corresponding to κ=0.15, κ=0.20, κ=0.25 and κ=0.35, respectively, were obtained by solving Equation (Equation 20), whereas the solid black curve corresponding to κ=0 was acquired through the numerical solution of Equation (Equation 18). Interestingly, the amplitude of the Landau damping declined drastically as κ increased, which demonstrated that the Landau damping was intensely depressed in the κ-deformed Kaniadakis distributed plasma system compared to the Maxwellian.

Next, to illustrate the effect of OAM on the Langmuir waves, we showed the dispersion variation and Landau damping with different values of the OAM parameter η while keeping κ fixed. The results are shown in Figure 3 and Figure 4. From Figure 3, it cqn be seen that the Langmuir wave dispersion shrank as the curves moved away from each other for higher wave number values. In addition, Figure 3 shows that increasing values of η resulted in a narrowing of Langmuir wave dispersion. In particular, the dispersion was very sensitive to the OAM parameter values that varied in the range 0.4≤η≤1 while there was little change beyond 1. This behavior was realized by looking at the term $1+1/\eta^{2}$ in Equation (Equation 14), which increased sharply in the range 0≤η≤1. In Figure 4, as η increased within the range 0.4≤η≤1, the Landau damping diminished. Conversely, when η increased within the range 1≤η≤∞, the Landau damping was enhanced at small and diminished at large wave numbers. Similarly, it was enhanced with the increased value of $1+1/\eta^{2}$ as the wave number gradually increased, which can be understood more naturally from Equation (Equation 18).

With the value of the OAM parameter η fixed, the dispersion variation and Landau damping of ion-acoustic waves with varying values of the deformation parameter κ are illustrated in Figure 5 and Figure 6. We solved Equation (Equation 23) numerically and plotted the Figure 5 with dashed blue, green, purple, and solid red curves corresponding to κ=0.15, κ=0.20, κ=0.25 and κ=0.35, respectively, while the solid black line corresponded to κ=0 according to the numerical solution to Equation (Equation 25).

From Figure 5, a very slight change was seen in the dispersion of ion-acoustic waves carrying OAM with increased κ, but the dispersion increased overall. The Landau damping of ion-acoustic waves with OAM was calculated primarily from Equation (Equation 26). In Figure 6, the dashed blue curve for κ=0.15 is clearly above the solid black line where κ=0 (Maxwellian case). Therefore, it was evident from the figure that the Landau damping rate of the non-Maxwellian ion-acoustic waves was smaller compared to the Maxwellian. The Figure 7 and Figure 8 show how the dispersion relation and the damping rate of ion-acoustic waves were affected by the presence of the OAM parameter η while keeping κ fixed. In Figure 7, the dispersion of ion-acoustic waves diminished as the OAM parameter η increased. From Figure 8, the Landau damping rate for ion-acoustic waves shifted to smaller negative values with increased η. In the case of the fixed deformation parameter κ, both dispersion and damping for ion-acoustic waves were suppressed by the OAM effect.

Physically, the thermal motion of particles gave rise to the dispersion of the longitudinal plasmon collective mode; the Landau damping was attributed to the resonance interaction between the plasma wave and the particles that had a velocity close to the phase velocity; therefore, the damping rate depended on the number of resonant particles. When the value of η was fixed, owing to Equations (14) and (23), the dispersion of Langmuir waves and ion-acoustic waves was viewed as being approximately proportional to $\sqrt{T_{eff}/m_{e}}$ and $\sqrt{T_{eff}/m_{i}}$, respectively. From Figure 1 of Ref. [53], as the κ increased, so did the effective temperature of the κ-deformed Kaniadakis distributed plasma system. Since $m_{e}\ll m_{i}$, the significant enhancement of Langmuir wave dispersion and the slight enhancement of ion-acoustic wave dispersion with increasing κ in Figure 1 and Figure 5 can be explained. Gougam showed that in the κ-deformed Kaniadakis distributed plasma system, the presence of high-energy states became more plausible as κ increased [57], which indicated an increase in the number of fast particles or a decrease in the number of slow particles with respect to the Maxwellian case. As a result, with increasing κ for both Langmuir and ion-acoustic waves, the phase velocity was skewed toward the trailing part of the distribution function; thus, the Landau damping amplitude diminished. When the κ value was fixed, the decline in η, namely, the rise in $1+1/\eta^{2}$, provoked the dissipation of Langmuir and ion-acoustic waves, the intensity of which was related to the magnitude of the OAM. This was due to the participation of partially resonant particles in the resonance of the wave OAM, which allowed a relative increase in the number of resonances obtaining energy from the wave, thereby leading to enhanced wave damping.

To summarize, electrostatic waves carrying OAM were first considered in a κ-deformed Kaniadakis plasma system by keeping kinetic theory in view. Statistically, the κ-deformed Kaniadakis distribution function was the product of superstatistics, which are perceived as a more generalized statistic in contrast to non-generalized and the traditional Boltzman–Gibbs statistics. Because it served the same purpose as the κ of the suprathermal distribution and the parameter q of the non-extensive distribution, the parameter κ of the κ-deformed Kaniadakis distribution function also measured the derivation from the Maxwellian distribution function, so the collective mode features of the associated plasma system were modified. The analytical expressions were derived for the dispersion relation and the Landau damping of both the Langmuir and ion-acoustic waves. It was shown that the presence of the κ-deformed Kaniadakis distribution function and OAM modified the propagation properties of the Langmuir and ion-acoustic waves. For both, the dispersion was enhanced with increased κ, while Landau damping was suppressed. Conversely, both the dispersion and Landau damping were depressed by the OAM effect. As expected, when κ=0 and η→∞, the results coincided with the straight propagating plane waves in a Maxwellian plasma system. It was also expected that the results of this study would offer more insight into the trapping and transportation of plasma particles and energy in a κ-deformed Kaniadakis distributed plasma system. In addition, the results may also provide a reference for studying its nonlinearity.

## Figures and Tables

**Figure 1 entropy-24-01211-f001:**
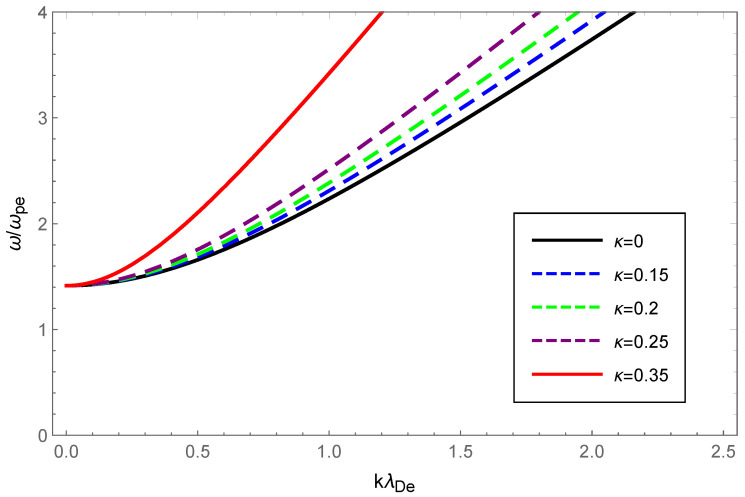
Variation in the normalized wave frequency $\omega /\omega_{\mathrm{pe}}$ of Langmuir waves with the normalized wave number $k\lambda_{De}$ for the deformation parameter κ=0 (black), κ=0.15 (dashed blue), κ=0.20 (dashed green), κ=0.25 (dashed purple) and κ=0.35 (solid red), respectively, with the OAM parameter η=1.

**Figure 2 entropy-24-01211-f002:**
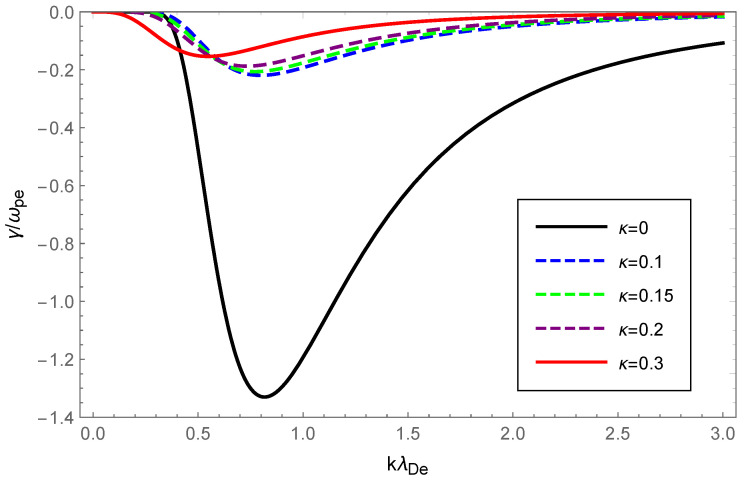
Variation of the Landau damping rate $\gamma /\omega_{\mathrm{pe}}$ of Langmuir waves with the normalized wave number $k\lambda_{De}$ for deformation parameter κ=0 (black), κ=0.15 (dashed blue), κ=0.2 (dashed green), κ=0.25 (dashed purple) and κ=0.35 (solid red), respectively, with the OAM parameter η=1.

**Figure 3 entropy-24-01211-f003:**
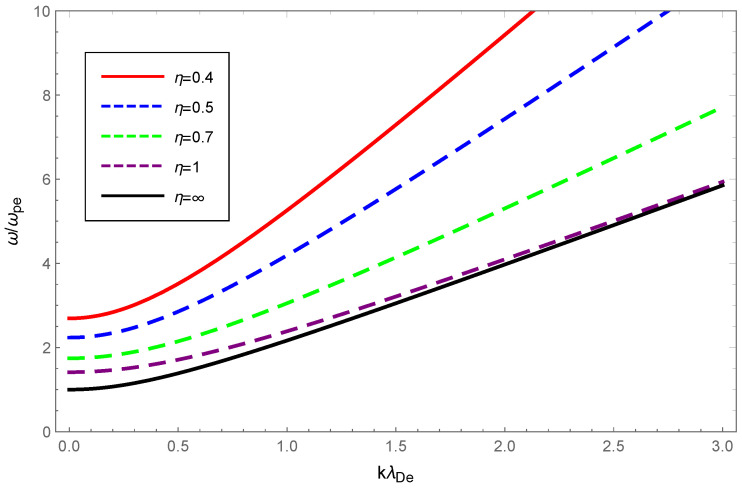
Variation of the normalized wave frequency $\omega /\omega_{\mathrm{pe}}$ of Langmuir waves with the normalized wave number $k\lambda_{De}$ for OAM parameter η=0.4 (solid red), η=0.5 (dashed blue), η=0.7 (dashed green), η=1 (dashed purple), and η=∞ (solid black), with the deformation parameter κ=0.2.

**Figure 4 entropy-24-01211-f004:**
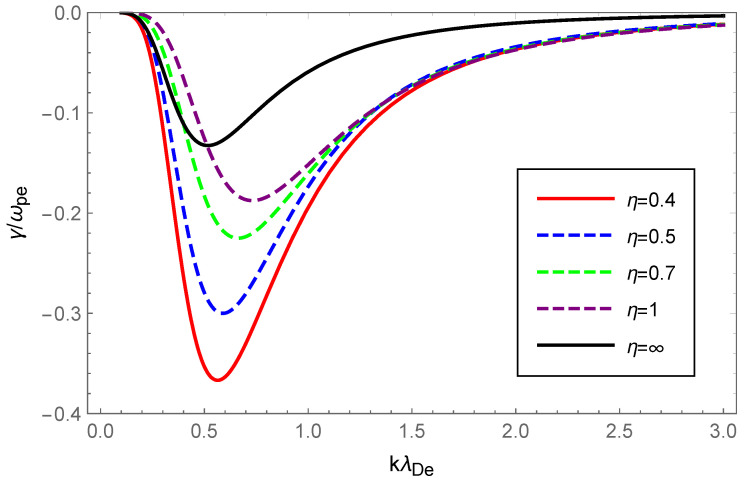
Variation of the Landau damping rate $\gamma /\omega_{\mathrm{pe}}$ of Langmuir waves with the normalized wave number $k\lambda_{De}$ for OAM parameter η=0.4 (solid red), η=0.5 (dashed blue), η=0.7 (dashed green), η=1 (dashed purple) and η=∞ (solid black) with the deformation parameter κ=0.2.

**Figure 5 entropy-24-01211-f005:**
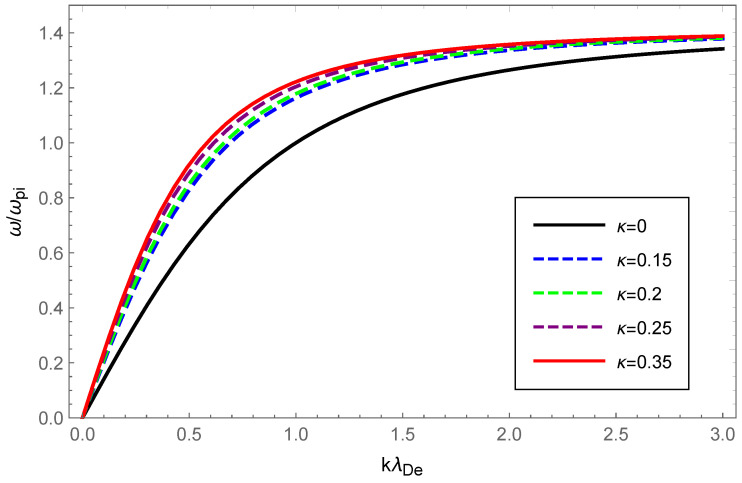
Variation of the normalized wave frequency $\omega /\omega_{\mathrm{pi}}$ of ion-acoustic waves with the normalized wave number $k\lambda_{De}$ for deformation parameter κ=0 (soild black), κ=0.15 (dashed blue), κ=0.2 (dashed green), κ=0.25 (dashed purple) and κ=0.35 (soild red) with OAM parameter η=1.

**Figure 6 entropy-24-01211-f006:**
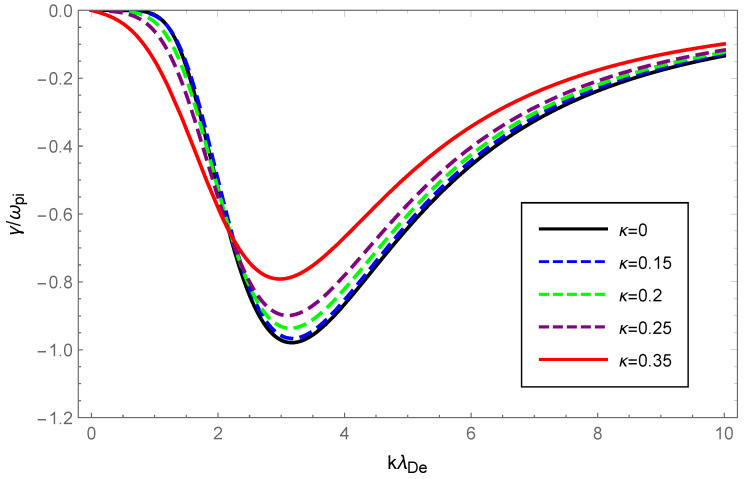
Variation of the Landau damping rate $\gamma /\omega_{\mathrm{pi}}$ of ion-acoustic waves with the normalized wave number $k\lambda_{De}$ for deformation parameter κ=0 (solid black), κ=0.15 (dashed blue), κ=0.2 (dashed green), κ=0.25 (dashed purple) and κ=0.35 (solid red) with OAM parameter η=1.

**Figure 7 entropy-24-01211-f007:**
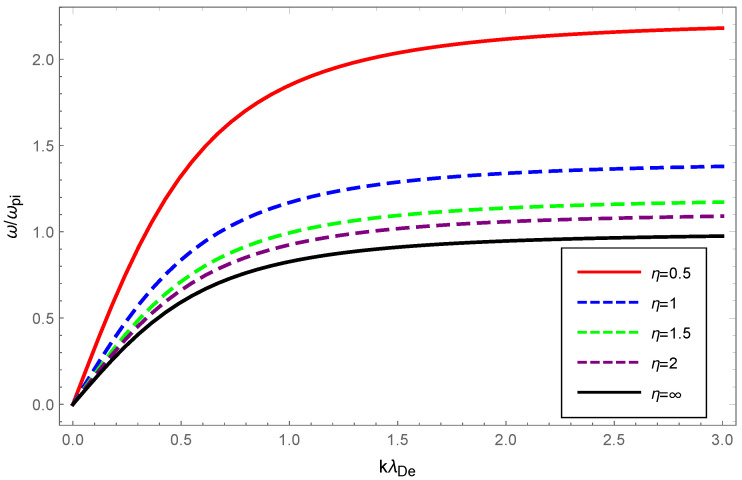
Variation of the normalized wave frequency $\omega /\omega_{\mathrm{pi}}$ of ion-acoustic waves with the normalized wave number $k\lambda_{De}$ for OAM parameter η=0.5 (solid red), η=1 (dashed blue), η=1.5 (dashed green), η=2 (dashed purple) and η=∞ (solid black) with deformation parameter κ=0.2.

**Figure 8 entropy-24-01211-f008:**
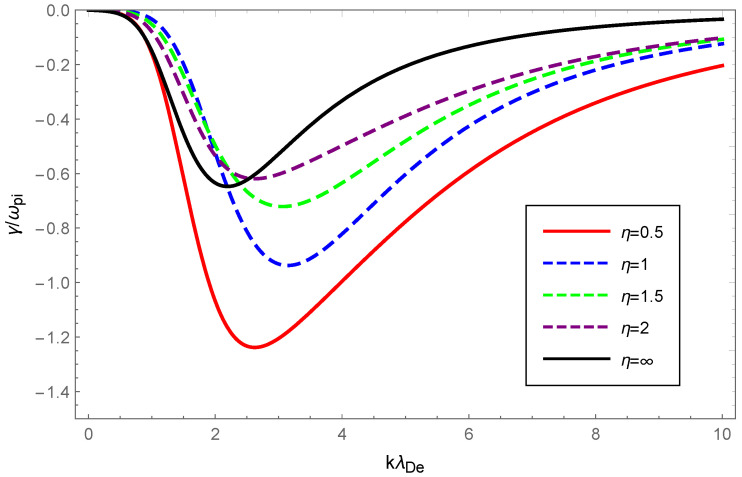
Variation of the Landau damping rate $\gamma /\omega_{\mathrm{pi}}$ of ion-acoustic waves with the normalized wave number $k\lambda_{De}$ for OAM parameter η=0.5 (solid red), η=1 (dashed blue), η=1.5 (dashed green), η=2 (dashd purple) and η=∞ (solid black) with deformation parameter κ=0.2.

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
