# Peer review of "The Longitudinal Plasma Modes of κ-Deformed Kaniadakis Distributed Plasmas Carrying Orbital Angular Momentum"

_entropy, 2022, doi:10.3390/e24091211_

Round 1

Reviewer 1 Report

The paper deals with the dispersions and Landau damping of the Langmuir waves and ion-acoustic waves carrying finite orbital angular momentum when the plasma system follows the κ-deformed Kaniadakis distribution. The Authors obtain the explicit dependence on the deformation parameter κ. of various physically meaningful quantities characterizing the system. In the classical limit where κ approaches the value zero, the already known formulas of Maxwettian plasmas are obtained. The present results give more insight into the trapping and transportation of plasma particles and energy. 

The paper is well-written and scientifically sound. Them of this work is of interest to the broad readership of the journal ENTROPY.  Therefore I recommend the publication of the paper in ENTROPY after the Authors take into account  the following minor technical suggestions/observations:

1) The acronym OAM in the title should be specified explicitly

2) The formula giving the effective temperature in the first line of page 3 could be numbered

3) In the title of section 3 correct the typing error in the name Landau

4) Eq. (17), formally is non-correct due to its big extension. The Authors could use the function exp_k(x)  to obtain smaller formula which can be typed better in one line only. The use of the function  exp_k(x) will help the reader in obtaining the classical  limit of the formula

5) Also Eq.(16) could be given in terms of the function  exp_k(x) 

6) Also Eq.(25) could be given in terms of the function  exp_k(x)

Reviewer 2 Report

This is a theoretical work on the Langmuir wave and ion-acoustic wave properties in the κ-deformed Kaniadakis distributions coupled with OAM. The authors focused on deriving the analytical expressions of the dispersion relations and the Landau damping rate of these two waves. Generally, I found no major issues in the equations. However, the scientific significance of this work is unclear.

The authors introduced the applications of OAM between Line 27 and 33, but they have nothing to do with plasmas. On-Line 57-58, the authors cited references about the OAM applications in plasmas. I would suggest the authors list those applications clearly in this paragraph. Also, these applications may not follow the κ distribution. The authors need to explain why the coupling of OAM and κ distribution are valuable to study. Specifically, please give some brain-storm examples of plasma engineering where OAM is working with κ distribution. Moreover, the authors are suggested to discuss how the results of this work can support those applications after showing all 8-figure results.

Another issue is the significance of deriving analytical relations. Nowadays, plasma applications are merged into engineering applications, each with specific geometry and properties. Therefore, theoretical studies are usually simulations, such as PIC. The simulations can contain spatial and temporal resolutions with inelastic collisions. A comparison with experimental results can also verify the simulation works. Therefore, considering this work is under the assumptions of collisionless plasma with no spatial and temporal resolution, in other words, at a local point and a short period of time, how informative of this work is questionable. Do the resulted models, such as Eq. (13) and Eq. (17) help those experimental researchers?  They may use simulation to improve their work rather than using the models derived in this work.

Other minor issues:

·      1. typo in section 3 title.

·    2. Move the abbreviation of Laguerre-Gaussian to the introduction where the first time this term appears.

Overall, I would like to suggest a revision and let the authors add more text to explain the above questions and correct issues.
